# Structural and Theoretical Study of Copper(II)-5-fluoro Uracil Acetate Coordination Compounds: Single-Crystal to Single-Crystal Transformation as Possible Humidity Sensor

**DOI:** 10.3390/polym15132827

**Published:** 2023-06-26

**Authors:** Verónica G. Vegas, Andrea García-Hernán, Fernando Aguilar-Galindo, Josefina Perles, Pilar Amo-Ochoa

**Affiliations:** 1Departamento de Química Inorgánica, Facultad de Ciencias, Universidad Autónoma de Madrid, 28049 Madrid, Spain; veronica.garciav@estudiante.uam.es (V.G.V.); andrea.garciah@uam.es (A.G.-H.); 2Departamento de Química, Facultad de Ciencias, Universidad Autónoma de Madrid, 28049 Madrid, Spain; fernando.aguilar-galindo@uam.es; 3Laboratorio de DRX Monocristal, Servicio Interdepartamental de Investigación, Universidad Autónoma de Madrid, 28049 Madrid, Spain; 4Institute for Advanced Research in Chemical Sciences (IAdChem), Universidad Autónoma de Madrid, 28049 Madrid, Spain

**Keywords:** copper, humidity sensors, coordination compounds, coordination polymers, theoretical calculations

## Abstract

This paper describes the synthesis and characterization of seven different copper(II) coordination compounds, as well as the formation of a protonated ligand involving all compounds from the same reaction. Their synthesis required hydrothermal conditions, causing the partial in situ transformation of 5-fluoro uracil-1-acetic acid (5-FUA) into an oxalate ion (ox), as well as the protonation of the 4,4′-bipyridine (bipy) ligand through a catalytic process resulting from the presence of Cu(II) within the reaction. These initial conditions allowed obtaining the new coordination compounds **[Cu_2_(5-FUA)_2_(ox)(bipy)]_n_·2n H_2_O (CP2)**, **[Cu(5-FUA)_2_(H_2_O)(bipy)]_n_·2n H_2_O (CP3)**, as well as the ionic pair **[(H_2_bipy)^+2^ 2NO_3_^−^] (1)**. The mother liquor evolved rapidly at room temperature and atmospheric pressure, due to the change in concentration of the initial reagents and the presence of the new chemical species generated in the reaction process, yielding CPs **[Cu(5-FUA)_2_(bipy)]_n_·3.5n H_2_O**, **[Cu_3_(ox)_3_(bipy)_4_]_n_** and **[Cu(ox)(bipy)]_n_**. The molecular compound **[Cu(5-FUA)_2_(H_2_O)_4_]·4H_2_O** (more thermodynamically stable) ended up in the mother liquor after filtration at longer reaction times at 25 °C and 1 atm., cohabiting in the medium with the other crystalline solids in different proportions. In addition, the evaporation of H_2_O caused the single-crystal to single-crystal transformation (SCSC) of **[Cu(5-FUA)_2_(H_2_O)(bipy)]_n_·2n H_2_O (CP3)** into **[Cu(5-FUA)_2_(bipy)]_n_·2n H_2_O (CP4)**. A theoretical study was performed to analyze the thermodynamic stability of the phases. The observed SCSC transformation also involved a perceptible color change, highlighting this compound as a possible water sensor.

## 1. Introduction

Coordination compounds (CCs) are formed by the self-assembly of several building blocks (usually metal ions and organic ligands) and have been known for more than 100 years [1]. Their research has not stopped growing in different fields (porous materials, cancer, bioimaging, photocatalysis, etc.) [2,3,4,5], with more than 10,000 annual publications in the last decade (Web of Science). They are usually obtained via direct synthesis in a single step, often under mild conditions [2,6,7].

The search for infinite structures (coordination polymers or CPs) is a subarea within CCs that has been of great interest, e.g., in the obtention of new semiconductor materials with selective molecular recognition or porous CPs (metal–organic frameworks, MOFs), among others [8,9,10,11,12,13].

These infinite structures are usually insoluble, and this aspect has hindered their structural characterization [14]. For this reason, the synthetic conditions have been modified using new synthesis routes under more extreme conditions (solvothermal synthesis) that facilitate the formation of quality crystals for their subsequent resolution via single-crystal X-ray diffraction [15]. The solvothermal reactions have, in turn, favored new unconventional coordination environments [16], in situ ligand transformations, redox processes of the metal center [17], and even reactions catalyzed by the presence of the metal ion [18], among others. The behavior of CCs under high pressure and temperature is an amazing research area that allows the creation of new materials with interesting properties [16,19,20].

On the other hand, a fundamental characteristic of CCs is that their bond is highly dynamic, which makes these materials very versatile since they respond to different stimuli [21,22,23]. The lability of the bonds allows interesting structural modifications in the same reaction medium, as a consequence of small differences in the medium temperature, or the reagents concentrations, among others [24]. These transformations can sometimes maintain the single-crystal structural integrity during the modification (single-crystal to single-crystal or SCSC transformations) and are also a very interesting source of knowledge [25,26].

SCSC transformations, in the case of CCs, are also favored by the lability of the coordination bond and are controlled by various factors such as the coordination preference of the metal ion. There are several types of external physical and chemical factors that favor or induce these transformations, such as light, pressure, cooling, heating [27,28], or the presence of gases [29]. Generally, the variations between the two structural types must be small; for example, the presence of gases can cause molecular substitution in the coordination environment of the metal center [30]. The breaking and rearrangement as a result of the new bond will change the crystalline structure of the final compound and, therefore, its properties [30].

For these reasons, the study of the structural modifications in the reaction medium is key and essential to understand the CC behavior, e.g., from the biochemical point of view (important in their possible use as pharmaceuticals or drug delivery) [31] or can be used in the design of new compounds with the desired properties. Indeed, SCSC studies may have applications in the manufacture of sensors, gas storage, etc.

In this sense, the study of low-cost humidity sensors, with easy handling (visual change) and high durability, is always desirable [32,33] and often essential in areas such as civil engineering and construction materials where it is important to determine the safety, reliability, and service life of large-scale public infrastructure highly sensitive to moisture and environmental factors that damage building structures [34]. The use of sensors will allow controlling their deterioration [35].

Along with technological advancements, many types of sensors have been used in construction engineering to provide a wide variety of data for management and maintenance. Currently, they are mostly applied to monitor indoor environments in order to enable comfortable environment management [36]. Indeed, according to McMullan [37], relative humidity should be kept between 30% and 70% to achieve thermal comfort and avoid condensation risks.

Therefore, in this work, in addition to the obtention, isolation, and characterization of eight different compounds starting from the same hydrothermal synthesis conditions using Cu(II), 4,4′-bypiridine (bipy), and 5-fluorouracil-1-acetic acid (**5-FUA**) as building blocks, a perceptible color change as a result of an SCSC transformation is also observed. This transformation is initiated by the decrease of water in the environment; consequently, taking advantage of this physical property, we study its possible behavior as a colorimetric humidity sensor. Additionally, we use theoretical calculations to understand the SCSC transformation observed.

## 2. Materials and Methods

All reagents and solvents were purchased from standard chemical suppliers and used as received: Cu(NO_3_)_2_·3H_2_O extra pure Scharlab (Barcelona, Spain) (CAS: 10031-43-3) and 4,4′-bypiridine 98%, ACROS Organics (Madrid, Spain)(CAS: 553-26-4), 5-fluorouracil 98%, ACROS Organics (Madrid, Spain) (CAS: 38771-21-1), and chloroacetic acid 99+% (CAS: 79-11-8). The synthesis of the 5-fluorouracil-1-acetic acid was performed in line with previous work [38].

**Infrared (FT-IR)** spectra were recorded on a PerkinElmer (Madrid, Spain) model 100 spectrophotometer using a PIKE (Cottonwood, WI, USA) Technologies MIRacle Single-Reflection Horizontal ATR Accessory from 4000–500 cm^−1^.

**Elemental analysis** was performed on an elementary microanalyzer LECO (Madrid, Spain) CHNS-932. It works with controlled doses of O_2_ and a combustion temperature of 1000 °C.

**Powder X-ray diffraction (PXRD)** were collected using a PANalytical (Malvern, UK) X’Pert PRO MPD θ/2θ secondary monochromator and detector with fast X’Celerator, which was used for general assays. The samples were analyzed with θ/2θ scanning. Theoretical X-ray powder diffraction patterns were calculated using Mercury software version 4.0.0 from the Crystallographic Cambridge Database Center (Cambridge, UK).

**Single-crystal X-ray diffraction (SCXRD)** was performed on suitable single crystals, which were isolated, coated with mineral oil, and mounted on MiTeGen MicroMounts(Ithaca, NY, USA). The samples were measured in a Bruker (Madison, WI, USA) D8 Kappa diffractometer (with APEX II area-detector system, equipped with graphite monochromated Mo Kα radiation (λ = 0.71073 Å). The substantial redundancy in data allowed empirical absorption corrections [39] to be applied using multiple measurements of symmetry-equivalent reflections. Raw intensity data frames were integrated with the SAINT [40] program, which was also used to apply corrections for Lorentz and polarization effects. The Bruker SHELXTL Software Package [41] was used for space group determination, structure solution, and refinement. The space group determination was based on a check of the Laue symmetry, and systematic absences were confirmed using the structure solution. The structures were solved by direct methods [42] completed with different Fourier syntheses and refined with full-matrix least-squares minimizing *w*(F_o_^2^–F_c_^2^)^2^. Weighted R factors (*w*R) and all goodness of fit S were based on F^2^; conventional R factors (R) were based on F. All scattering factors and anomalous dispersion factors were contained in the SHELXTL 6.10 program library. All non-hydrogen atoms were refined with anisotropic displacement parameters. Hydrogen atoms were placed at calculated positions except the ones from water molecules included in the models, which were located in the electron density maps.

The crystal structures of the compounds **[(H_2_bipy)^+2^ 2NO_3_^−^] (1), CP2, CP3,** and **CP4** were deposited in the CSD (https://www.ccdc.cam.ac.uk, accessed on 18 June 2023) with CCDC numbers 2243124, 2243125, 2243126, and 2243127, respectively; they can be retrieved free of charge from the Cambridge Crystallographic Data Center via www.ccdc.cam.ac.uk/data_request/cif (accessed on 18 June 2023) (Appendix A).


**Synthesis of [(H_2_bipy)^+2^ 2NO_3_^−^] (1), [Cu_2_(5-FUA)_2_(ox)(bipy)]_n_·2n H_2_O (CP2), [Cu(5-FUA)_2_(H_2_O)(bipy)]_n_·2n H_2_O (CP3)**


5-Fluorouracil-1-acetic acid (**5-FUA**) (75.5 mg, 0.41 mmol), Cu(NO_3_)_2_·3H_2_O (54.6 mg, 0.205 mmol), and bipy (0.125 g, 0.8 mmol) were mixed in 9 mL of MilliQ water (pHi = 2.3) in a glass hydrothermal (25 mL) reactor and magnetically stirred (900 rpm) for 10 min at room temperature. During this time, a color change from light to dark blue was observed. Then, the mixture was heated at 120 °C for 48 h, followed by a cooling ramp at ca. 0.066 °C/min. After that (pH_f_ = 2.75), a mixture of compounds **CP2**, **CP3**, and **1** (Appendix A) in different proportions was obtained. The mixture was filtered off and manually separated under optical microscopy. Turquoise crystals (**CP2**) were suitable for single-crystal X-ray diffraction studies (Appendix A), (75% yield based on Cu). Analytically calculated (found) for C_12_H_13_CuFN_3_O_9_: C, 37.1% (36.8%); H, 2.1% (2.3%); N, 10.9% (10.5%). Characteristic IR bands (cm^−1^): 3065 (w), 1672(s), 1595 (s), 1400 (m), 1372 (m), 1242 (m) 1076 (w), 985 (m), and 824 (m) (Appendix A). Blue crystals suitable for single-crystal X-ray diffraction studies of **CP3** (Appendix A) were also isolated (yield 3% based on Cu). Characteristic IR bands (cm^−1^): 3460 (w), 1698 (s), 1600 (s), 1407 (m), 1302 (m), 1236 (s), and 823 (w). Yellow crystals suitable for single-crystal X-ray diffraction studies ([(H_2_bipy)^+2^ 2.NO_3_^−^] **(1)**) (Appendix A) were also isolated (yield 7% based on Cu). Characteristic IR bands (cm^−1^): 3062 (m), 2938 (w), 1720 (w), 1641 (s), 1430 (s), 1347 (m), 1239 (s), 867 (m), and 806 (s) (Appendix A).


**Single-crystal to single-crystal transformation of compound [Cu(5-FUA)_2_(H_2_O)(bipy)]_n_·2n H_2_O (CP3) into [Cu(5-FUA)_2_(bipy)]_n_·2n H_2_O (CP4)**


Once compound **CP3** was separated from the mother liquor in ambient conditions, a color change from dark blue to violet occurred, indicative of a single-crystal to single-crystal transformation. The resulting crystals, with formula **CP4** were suitable for single-crystal X-ray diffraction structure solution (Appendix A). Analytically calculated (found) for C_11_H_10_Cu_0.5_FN_3_O_5_: C, 40.8% (41.1%); H, 3.4% (2.9%); N, 12.9% (12.6%).


**Humidity sensor study**


To measure the percentage of moisture at which the violet compound **CP4** changed its color reversibly to blue **CP3**, **c**rystals of the former compound, together with a hygrometer, were placed into a dry box with an initial environmental humidity percentage between 25% and 35% (the experiment was carried out three times to ensure reproducibility). Inside the dry box, a beaker with water on a heating plate was also placed to increase the environmental humidity, observing the change in color to blue at 70% relative humidity.


**Theoretical calculations**


All the calculations were carried out using the Vienna Ab initio Simulation Package (VASP) in the framework of the density functional theory (DFT), with the OPTPBE functional, which takes into account weak interactions (i.e., van der Waals forces). Periodic boundary conditions (PBC) were included in the calculation, allowing us a correct representation of the crystalline nature of the materials.

The electron density was expanded on a plane-wave basis, which is the ideal basis set to deal with periodic systems, such as those studied in this work. We included plane waves up to a cutoff energy of 450 eV, while the interactions between nuclei and electrons were described with the projector augmented wave (PAW) pseudopotentials, as provided in the VASP database. Reciprocal space was sampled with the Γ-point.

We used as the convergence criterium 10^−5^ eV for the electron density. During the optimizations, we imposed all the Hellmann–Feynman forces to be lower than 0.01 eV/Å.

## 3. Results and Discussion

### Structural Characterization

The reaction carried out in water under hydrothermal conditions (120 °C for 48 h) in a 1:4:2 stoichiometric ratio, (copper(II) nitrate trihydrate, 4,4-bipyridine (**bipy**), and 5-fluorouracil-1-acetic acid (**5-FUA)**) at pH around 2.3 produced several crystalline phases, some of them containing oxalate ((C_2_O_4_)^2−^ = **ox**), acting as a bridging ligand. This experimental observation can be rationalized, taking into account previous reports in which the use of solvothermal conditions, in combination with acid pH values and the presence of copper(II) as a catalyst and reducing agent, produced an in situ decarboxylation of carboxyl groups. Thus, the carboxylic acid group became CO_2_, which was promoted to (C_2_O_4_)^2−^ in the presence of Cu^2+^, via a reduction reaction [43,44,45].

The careful observation at different times of the reaction mixture inside of the reactor (Table 1) and the mother liquor after filtration (Table 2), and the manual separation of the appearing single crystals allowed us to identify eight different crystalline phases, and to study the transformations between them. Interestingly, all the copper(II) compounds were pleochroic, showing important variations in their color depending on the orientation (Figure 1), which made more challenging the visual identification and separation of the phases.

As described in the synthesis section, as soon as the hydrothermal reaction finished, two types of blue crystals, together with aggregates of yellow solid, were observed inside the reactor (Figure 2a and Appendix A). Some of the mother liquor was separated from the reaction mixture and left at room temperature.

In the isolated mixture, we observed that the yellow solid was composed of very small yellow crystals that partially dissolved if left in the mother liquor (Figure 2b,c). The IR spectrum of compound **1** showed bands at 3062 and 2938 cm^−1^ corresponding to the ν(C–H) stretching vibrations, as well as bands around 1641 and 806 cm^−1^ attributed to out-of-plane ν(C–H) stretching vibrations of the pyridine ring [46]. In addition, the vibrational peak for N=O stretching appeared at 1720 cm^−1^. The NO_3_^−^ asymmetric and symmetric stretching vibrations appeared at 1347 and 867 cm^−1^, respectively, supporting the presence of NO_3_^−^ as a counterion (Appendix A) [47]. Finally, single crystals of adequate size for SCXRD analysis were obtained (Appendix A). It was found to be a new polymorph of the ionic solid with formula **[(H_2_bipy)^+2^ 2 NO_3_^−^]** and did not contain any copper ions.

One of the blue phases appeared as deep blue/turquoise crystals in the shape of long needles or ribbons, mostly forming aggregates (Figure 2d). The IR spectrum of this compound showed bands indicative of the presence of 5-FUA, oxalate, and bipyridine, displaced with respect to the starting ligands, indicating coordination to the metal center. The asymmetric and symmetric νCOO^−^ appeared at 1681 and 1401 cm^−1^, respectively [48]. In addition, the bands corresponding to ν_as_ and ν_s_ of C=O could be assigned at 1654 and 1401 cm^−1^, respectively, indicative of oxalate acting as a bridging ligand. The vibrations corresponding to the νCH of the aromatic ring were at 1592 and 1372 cm^−1^. Single crystals were isolated from the reaction mixture and analyzed by SCXRD, resulting in a new 2D polymer with formula **[Cu_2_(5-FUA)_2_(ox)(bipy)]_n_·2nH_2_O (CP2)** (Figure 3 and Appendix A). The structure could be described as wavy layers parallel to the (110) plane where the metal atoms were joined by the three ligands (5-FUA, ox, and bipy) acting as ditopic connectors. The oxalate ligands linked copper atoms along the [110] direction in a µ-1𝜅^2^O,O′:2𝜅^2^O″,O′′′ mode, alternating with the 4,4′-bipy ligands (as ditopic µ-1𝜅N:2𝜅N′), that connected the metals in the same direction. Lastly, the carboxylate group in the 5-FUA ligands acted as a bridge in the [100] direction to complete the CuO_5_N coordination environment. This resulted in an arrangement within the layers of pairs of Cu–O–Cu–O chains parallel to the [100] direction, where the copper atoms were located at a distance of 4.734 Å. There were numerous hydrogen bonds present, not only intralayer, but also between the sheets, involving the interstitial water molecules located between the 2D polymers (Figure 3). In addition, its elemental analysis matched the chemical formula obtained and its powder X-ray diffractogram showed the presence of a single crystalline phase coincident with the theoretical X-ray diffractogram obtained from single crystal X-ray diffraction (Appendix A).

If these crystals were left in the mother liquor, they began to show cracks after a few days, losing crystallinity and partially redissolving to yield **[Cu(ox)(bipy)]_n_** and **[Cu_3_(ox)_3_(bipy)_4_]_n_** (Table 1 and Appendix A).

Another type of blue crystal, plate-shaped and dichroic (from intense blue to cyan), appeared as stacked aggregates (Figure 2e). This crystalline phase constituted a new 1D polymer that did not contain the oxalate ligand, with formula **[Cu(5-FUA)_2_(H_2_O)(bipy)]_n_·2nH_2_O** (**CP3**, Figure 4, top). In this structure, only the bipyridine ligand acted as a connector along the growth direction of the polymer (parallel to [010]) in a ditopic µ-1𝜅N:2𝜅N′ fashion, and all of 5-FUA ligands were terminal, half of them coordinating to a copper atom in a 1𝜅^2^O,O′ chelating mode and the other half as monodentate 1𝜅O (Appendix A). There was also a coordinated water molecule per copper atom completing the distorted octahedral CuO_4_N_2_ environment. As with **[Cu_2_(5-FUA)_2_(ox)(bipy)]_n_·2nH_2_O** (**CP2**), there were numerous hydrogen bonds present between the polymers, involving the interstitial water molecules located between them (Appendix A).

**CP3** was obtained with a low yield. The powder X-ray diffractogram of the sample allowed us to conclude that we had a single crystalline phase coincident with the theoretical X-ray diffractogram obtained from single-crystal X-ray diffraction (Appendix A).

A small violet plate appeared after a few hours, but only in the upper parts of the walls of the reactor that were not covered in the solution, representing a new monodimensional CP with formula **[Cu(5-FUA)_2_(bipy)]_n_·2nH_2_O** (**CP4**, Figure 4, bottom) that could also be obtained when phase **[Cu(5-FUA)_2_(H_2_O)(bipy)]_n_·2nH_2_O (CP3)** was exposed to the ambient atmosphere, constituting a reversible single-crystal to single-crystal transformation.

When exposed to humidity or submerged in the mother liquor, the violet crystals quickly reverted to blue **[Cu(5-FUA)_2_(H_2_O)(bipy)]_n_·2nH_2_O (CP3)** (Figure 5). To confirm that the crystallinity of the solid was preserved, and to prove the SCSC nature of the conversion, the same single crystal previously used for the structure solution of **CP3** was left in ambient conditions until the color change was complete. After 2 days, it was submitted to a new SCXRD experiment, and the structure of **CP4** was solved from that data collection (Appendix A).

The main difference between these two 1D polymers was the absence of the coordinated water molecule present in the blue crystals, and the subsequent approximation of the previously non-coordinated carboxylic oxygen atom from half of the 5-FUA ligands. To achieve this transformation, the coordinated water molecule was lost (Cu–O distance 2.341 Å), and the previously non-coordinated carboxylic oxygen atom from half of the 5-FUA ligands approximated to the nearest metal center in order to complete the CuN_2_O_4_ coordination environment, changing its distance to the metal from 3.152 to 2.608 Å. This small rearrangement did not substantially modify the overall crystal structure, allowing the reversibility of the SCSC transformation. The fact that the coordinated water molecule was lost (causing the change in color) and the interstitial molecules remained was previously observed in our research [49]. This could be explained by the fact that these interstitial water molecules were involved in numerous hydrogen bonds (see Appendix A), and the energy needed to remove them was larger than that required to extract the molecule that is within the coordination sphere.

This transformation, accompanied by an important change in color, allowed us to postulate the possible use of this compound as a sensor of environmental humidity. To determine the degree of sensitivity, compound **[Cu(5-FUA)_2_(bipy)]_n_·2nH_2_O (CP4)** (violet) was placed in a dry box with relative humidity control (Figure 6). The experiment showed that, at humidity levels around 70%, the transformation to the compound **[Cu(5-FUA)_2_(H_2_O)(bipy)]_n_·2nH_2_O (CP3)** (blue) occurred. The process was reversible and reproducible. Indeed, the experiment was repeated up to three times to check the reversibility at different temperatures (20, 30, and 40 °C). **CP3** took 25 to 30 min to fully convert to **CP4** always at the same humidity percentage. In addition, the **CP4** elemental analysis obtained after the transformation matched with the chemical formula, and its X-ray powder diffractogram confirmed the obtention of a single crystalline phase coinciding with the theoretical X-ray diffractogram obtained by single-crystal X-ray diffraction (Appendix A).

Bearing in mind the importance of humidity in the conservation of buildings, in the comfort of homes, where the current regulations for thermal installations establish that the relative humidity inside buildings must be around 45–75% both compounds could serve as low-cost, real-time, in situ visualization and easy-to-handle humidity colorimetric sensors [36,50,51,52,53].

DFT calculations allowed us to understand the reversibility of this process. Using a large unit cell (*n* = 8, optimized structures are shown in Appendix A), **[Cu(5-FUA)_2_(H_2_O)(bipy)]_n_·2nH_2_O** was found to be 4.86 eV more stable than **[Cu(5-FUA)_2_(bipy)]_n_·2nH_2_O** + 8 isolated water molecules (see Table 3). However, if these molecules presented intermolecular interactions (e.g., in liquid phase), the relative stability reverted, and **[Cu(5-FUA)_2_(bipy)]_n_·2n H_2_O** became 0.11 eV more stable. Therefore, the observed crystalline phases were determined not only by their intrinsic properties, but also by the medium, since external conditions could shift the equilibrium.

After several days, two different crystal phases appeared inside of the reactor: small pale blue needles of **[Cu_3_(ox)_3_(bipy)_4_]_n_** and blue crystals of **[Cu(ox)(bipy)]_n_** (Appendix A).

The blue needles (pale blue/turquoise) were isolated and identified as the 2D coordination polymer **[Cu_3_(ox)_3_(bipy)_4_]_n_**, previously reported by Castillo et al. [54] and deposited in the CSD with code IJASAN. This phase could only be found at the early stages, spontaneously dissolving in the mother liquor unless filtered and separated from the mixture. The polymer consisted of copper(II) centers coordinated by oxalato ligands acting as µ-1𝜅^2^O,O′:2𝜅^2^O″,O′′′ bridges in the [010] direction to form chains. These chains were linked by ditopic µ-1𝜅N:2𝜅N’ 4,4′-bipy ligands, alternating with terminal bipy moieties, resulting in two-dimensional layers parallel to the (102) plane. Each metal atom was connected to three adjacent copper atoms (two in the same row and another from a neighboring one), presenting a distorted octahedral CuO_4_N_2_ coordination environment.

Another bidimensional coordination polymer appeared as dichroic pale green/intense blue crystals, in the shape of elongated prisms with a rhombic section. This phase started as very small needles that underwent an Ostwald ripening process and coalesced to yield larger crystals. They were identified as the compound **[Cu(ox)(bipy)]_n_**, also reported in the literature [55] and deposited in the CSD with code UDOGEA01. The crystal structure of this polymer consisted of layers, in this case parallel to the (011) plane, containing copper(II) centers in distorted octahedral CuO_4_N_2_ coordination environments. The oxalate ligands also acted as µ-1𝜅^2^O,O′:2𝜅^2^O″,O′′′ bridges, here in the [001] direction, to yield rows linked in the [010] direction by the 4,4′-bipy ligands; in this case, all of them were ditopic µ-1𝜅N:2𝜅N′. Each metal atom was connected to four adjacent copper atoms (two in the same row and two from the neighboring ones).

On the other hand, if the mother liquor was separated from the reaction mixture, only the monodimensional polymers **[Cu(5-FUA)_2_(H_2_O)(bipy)]_n_·2nH_2_O (CP3)** (as well as the dehydrated **[Cu(5-FUA)_2_(bipy)]_n_·2nH_2_O (CP4)** when the walls of the recipient were no longer covered in the solution) and **[Cu(5-FUA)_2_(bipy)]_n_·3.5nH_2_O** were found, together with **[Cu(5-FUA)_2_(H_2_O)_4_]·4H_2_O** (Figure 7 and Appendix A).

The structure of the purple plate-shaped crystals of polymeric **Cu(5-FUA)_2_(bipy)]_n_·3.5nH_2_O** was recently reported by our group [31] and deposited in the CSD with code ECAQEI. It is a ladder-type 1D coordination polymer with formula **[Cu(5-FUA)_2_(bipy)]_n_·3.5nH_2_O**, where the bipyridine acts as a µ-1𝜅N:2𝜅N′ connector. Half of the 5-FUA ligands act as bridges in a µ-1:2𝜅O fashion (Cu···Cu distance is 3.396 Å), and the other half are monodentate 1𝜅O.

The last crystalline phase identified appeared as large dichroic (turquoise/deep blue) prisms and was found to be a molecular coordination compound with formula **[Cu(5-FUA)_2_(H_2_O)_4_]·4H_2_O** (Appendix A), previously reported [56] and deposited in the CSD with code XAXMEO. In this compound, the copper atom adopted a CuO_6_ octahedral environment where the four equatorial positions were occupied by water molecules and the two axial ones were occupied by carboxylic oxygen atoms from terminal 5-FUA ligands (1𝜅O).

## 4. Conclusions

The present study unveiled the remarkable uniqueness of coordination compounds, highlighting their structural versatility. The variations in synthetic conditions and the dynamic nature of coordination bonds confirmed the boundless architectural possibilities and diverse properties inherent in these materials. Moreover, by employing more aggressive synthesis methods, we successfully obtained novel coordination environments and in situ transformations of the building blocks, often facilitated by the catalytic influence of the metal cation center. The low energy barriers between different environments facilitated the evolutionary behavior of these species over time.

This research expands the scope of structural possibilities explored thus far, employing Cu(II), 5-fluoro uracil-1-acetate, and 4,4-bipyridine. Specifically, the synthesis of the **[Cu(5-FUA)_2_(H_2_O)(bipy)]_n_·2nH_2_O** compound in its solid state, along with its subsequent transformation to **[Cu(5-FUA)_2_(bipy)]_n_·2nH_2_O** through a single-crystal to single-crystal (SCSC) process, demonstrates a pronounced color change. This phenomenon holds implications for potential applications, such as humidity sensing. Theoretical investigations have indicated that, although the compound with the coordinated water molecule exhibits greater energetic stability, the loss of the water molecule coordinated to the metallic center is favored by the formation of intermolecular bonds between them. Notably, an equilibrium between the two species was observed in the presence of approximately 70% humidity.

In summary, the research surpasses the existing state of the art by revealing novel synthetic pathways and transformations, leading to diverse architectural possibilities. The findings, particularly the SCSC transformation and the associated color change, hold useful for applications such as humidity sensing. Theoretical insights further illuminate the delicate balance between stability and intermolecular interactions, highlighting the complex nature of these compounds.

## Figures and Tables

**Figure 1 polymers-15-02827-f001:**
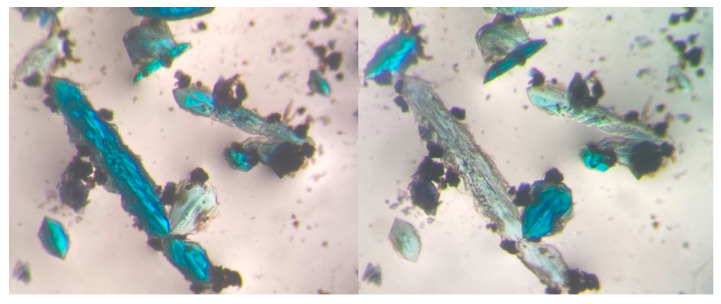
Color change under polarized light of pleochroic crystals from compound **[Cu_2_(ox)(bipy)]_n_**.

**Figure 2 polymers-15-02827-f002:**
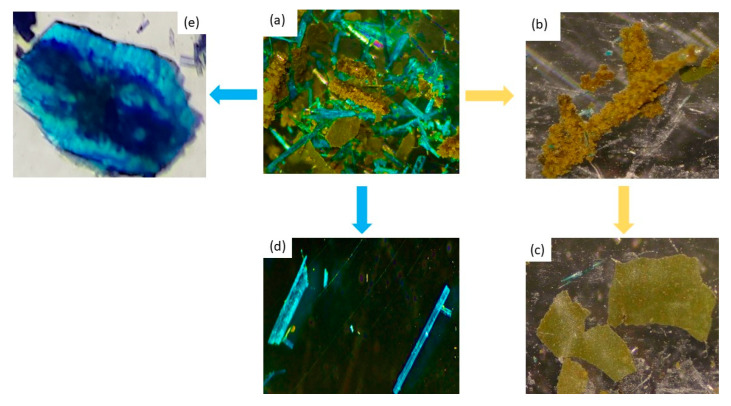
Initial mixture of yellow solid, together with pale-blue needle-shaped crystals (**a**). Polycrystalline yellow samples of **[(H_2_bipy)^+2^ 2.NO_3_^−^] (1)** (**b**,**c**). Pale blue needle-shaped crystals of **[Cu_2_(5-FUA)_2_(ox)(bipy)]_n_·2nH_2_O (CP2)**, (**d**), and **[Cu(5-FUA)_2_(H_2_O)(bipy)]_n_·2nH_2_O (CP3)** (**e**).

**Figure 3 polymers-15-02827-f003:**
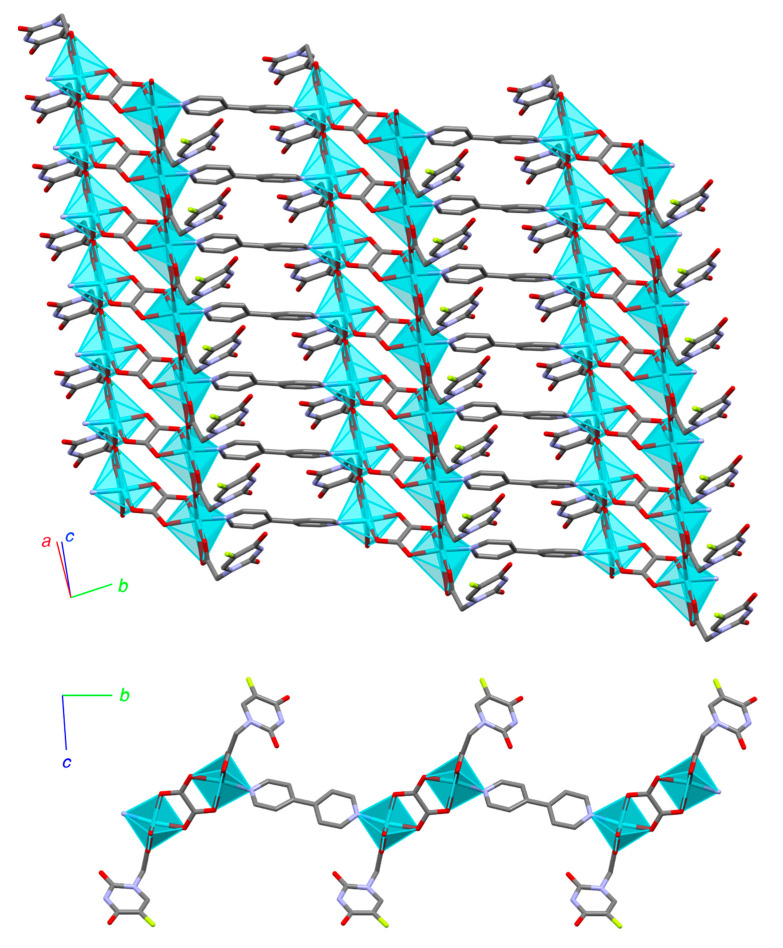
Layer of **[Cu_2_(5-FUA)_2_(ox)(bipy)]_n_ 2nH_2_O (CP2)**: (**top**) zenithal view of the layer; (**bottom**) lateral view, both with crystallographic axes. Cyan polyhedra represent the copper coordination environments.

**Figure 4 polymers-15-02827-f004:**
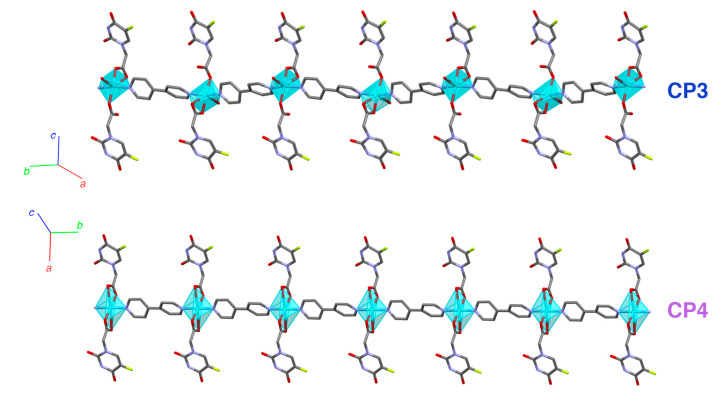
Coordination polymeric chains of compounds **CP3** (**top**) and **CP4** (**bottom**), both with crystallographic axes. Cyan polyhedra represent the copper coordination environments.

**Figure 5 polymers-15-02827-f005:**
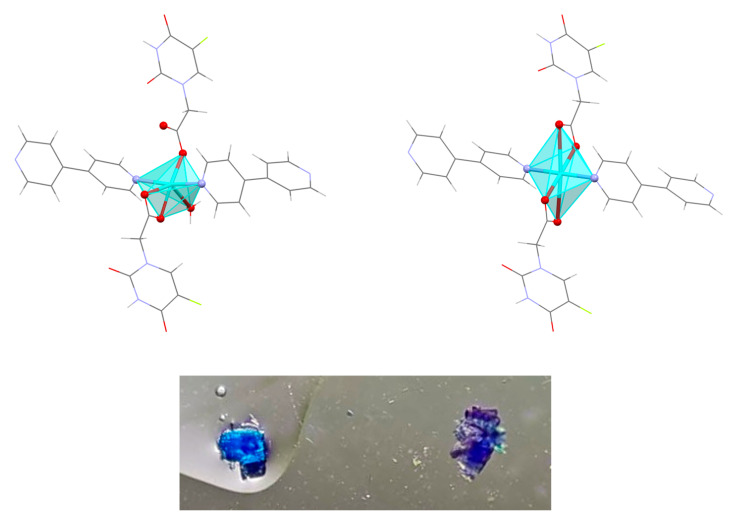
Reversible single-crystal to single-crystal transformation of **CP3** (**left**, in a drop of water) into **CP4** (**right**, exposed to air). Cyan polyhedra represent the copper coordination environments.

**Figure 6 polymers-15-02827-f006:**
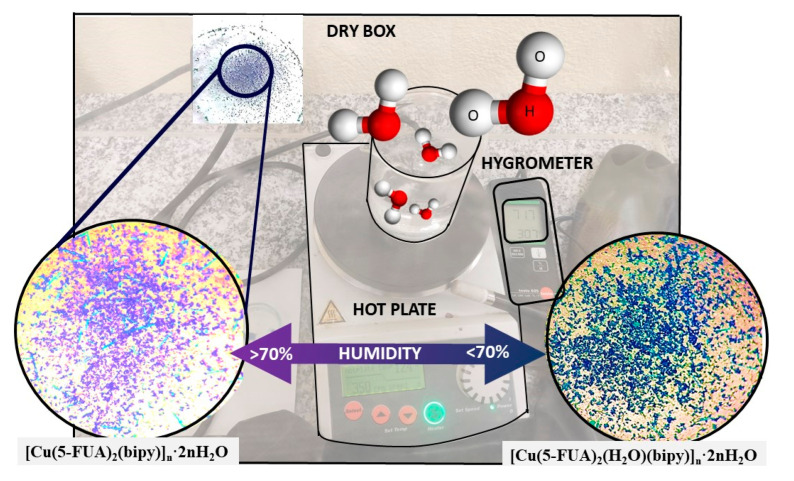
Diagram of the experiment carried out to measure the degree of humidity and the color change of **[Cu(5-FUA)_2_(bipy)]_n_·2nH_2_O (CP4)** (violet) and **[Cu(5-FUA)_2_(H_2_O)(bipy)]_n_·2nH_2_O (CP3)** (blue) compounds.

**Figure 7 polymers-15-02827-f007:**
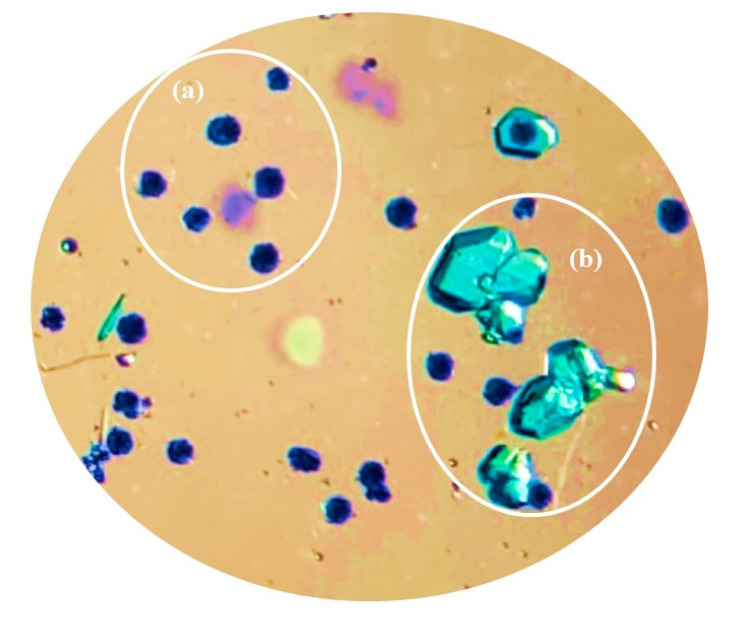
Optical image of **[Cu(5-FUA)_2_(bipy)]_n_·3.5nH_2_O** (**a**) and **[Cu(5-FUA)_2_(H_2_O)_4_]·4H_2_O** (**b**) in the separated mother liquor.

**Table 1 polymers-15-02827-t001:** Evolution of the coordination compounds (CCs) inside of the hydrothermal reactor from time 0 to 1 month.

Time	0 (Initial)	½ Day	6 Days	30 Days
CCs and concentration variation	[(H_2_bipy)^+2^ 2NO_3_^−^] (1)	Stable	Decrease	Decrease
	[Cu_2_(5-FUA)_2_(ox)(bipy)]_n_ ·2n H_2_O (CP2)	Stable	Decrease	[Cu_3_(ox)_3_(bipy)_4_]_n_ [Cu(ox)(bipy)]_n_
	[Cu(5-FUA)_2_(H_2_O)(bipy)]_n_ ·2n H_2_O (CP3)	Increase	Increase	Decrease
		[Cu(5-FUA)_2_(bipy)]_n_ ·3.5n H_2_O	Increase	Increase

**Table 2 polymers-15-02827-t002:** Evolution of the coordination compounds (CCs) in the mother liquor after filtration from time 0 to 6 days.

Time	½ Day	6 Days	30 Days
CCs and concentration variation		[Cu(5-FUA)_2_(H_2_O)_4_]·4H_2_O	Increase
	[Cu(5-FUA)_2_(H_2_O)(bipy)]_n_ ·2n H_2_O (CP3)	Increase	Decrease
	[Cu(5-FUA)_2_(bipy)]_n_ ·3.5n H_2_O	Increase	Increase

**Table 3 polymers-15-02827-t003:** Relative energy of the two phases with potential application as humidity sensor.

Compound	Relative Energy (eV)
[Cu(5-FUA)_2_(H_2_O)(bipy)]_n_·2nH_2_O	0.00
[Cu(5-FUA)_2_(bipy)]_n_·2nH_2_O + 8H_2_O (isolated)	4.86
[Cu(5-FUA)_2_(bipy)]_n_·2nH_2_O + 8H_2_O (condensed phase)	−0.11

## Data Availability

The data presented in this study are available on request from the corresponding author. The data are not publicly available due to privacy.

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
