# Peer review of "Structural and Theoretical Study of Copper(II)-5-fluoro Uracil Acetate Coordination Compounds: Single-Crystal to Single-Crystal Transformation as Possible Humidity Sensor"

_polymers, 2023, doi:10.3390/polym15132827_

Round 1

Reviewer 1 Report

This work reports the interesting crystal structure transformation of copper(II)-5-fluoro uracil acetate coordination compounds. The crystallographic data are diligently studied, and the results obtained are very impressive. The figures in the manuscript are beautiful and have high resolution. The proposed application is also found. I recommend publishing this manuscript after minor revisions as follows:

1.       Crystal structure solution and refinements

-          According to the CheckCIF reports for data blocks 03168_150K_II, 03168_III, and IV_03168_2021, the alert “PLAT007_ALERT_5_G Number of Unrefined Donor-H Atoms” is observed. The authors must delete H that is bonded to the heteroatom (N) that was previously refined by the riding model. Then, the authors must refine it again by assigning the Q peak nearby as the H atom so that this alert G will be wiped out. If the H is refined by this method, the N-H bond length and related bond angles will have standard uncertainties. Then please also correct Table S11 and other related tables.

2.       Materials and Methods

-          Please add software references for the following software and database: SADABS, SAINT, SHELXS, SHELXT, and CSD.

3.       Results and Discussion

-          Please add crystallographic axes in Figures 3 and 4 to clearly show the direction of the polymeric chain propagation as described in the text.

-          Nitrogen sorption isotherms are recommended to study as these materials will be intentionally used as sensors.

4.       Supporting information

-          The formats (fonts, line widths, etc.) of XRD diffractograms and FTIR spectra are inconsistent.

-          Please give the details of the peak assignments for FTIR spectra reported in the supporting information. 

Minor editing of English language is required

Author Response

Point-by-point response to the reviewers

Reviewer 1

This work reports the interesting crystal structure transformation of copper(II)-5-fluoro uracil acetate coordination compounds. The crystallographic data are diligently studied, and the results obtained are very impressive. The figures in the manuscript are beautiful and have high resolution. The proposed application is also found. I recommend publishing this manuscript after minor revisions as follows:

Crystal structure solution and refinements

Q1-According to the CheckCIF reports for data blocks 03168_150K_II, 03168_III, and IV_03168_2021, the alert “PLAT007_ALERT_5_G Number of Unrefined Donor-H Atoms” is observed. The authors must delete H that is bonded to the heteroatom (N) that was previously refined by the riding model. Then, the authors must refine it again by assigning the Q peak nearby as the H atom so that this alert G will be wiped out. If the H is refined by this method, the N-H bond length and related bond angles will have standard uncertainties. Then please also correct Table S11 and other related tables.

A1. We thank the referee for their helpful suggestion. The changes asked have been made (location of the hydrogen atoms in the electron density maps and refinement of the parameters, modification in all of the tables needed), and the structures have been redeposited in the CSD.

Materials and Methods

-Q2. Please add software references for the following software and database: SADABS, SAINT, SHELXS, SHELXT, and CSD.

A2. We have added all of the references in the manuscript.

Results and Discussion

Q3-Please add crystallographic axes in Figures 3 and 4 to clearly show the direction of the polymeric chain propagation as described in the text.

A3. We modified the figures as requested.

Q4-Nitrogen sorption isotherms are recommended to study as these materials will be intentionally used as sensors.

A4. We appreciate reviewer 1's suggestion, however, the coordination polymers presented in this work do not present porosity, so it is not necessary to carry out nitrogen adsorption isotherms. Their capacity as sensors is determined by a color change as a consequence of the loss or entry of water molecules in the coordination environment.

Supporting information

Q5. The formats (fonts, line widths, etc.) of XRD diffractograms and FTIR spectra are inconsistent.

A5. According to the reviewer, the FTIR and XRD-diffractograms figures have been put in the same format.

Q6. Please give the details of the peak assignments for FTIR spectra reported in the supporting information. 

A6. Following the reviewer's indications, we have proceeded to add the peak details in the FTIR spectra of the supporting information.

Reviewer 2 Report

In this manuscript, the authors isolate eight compounds from a hydrothermal synthesis system and characterize their structures. This manuscript is generally systematic. Four of the isolated compounds are new, and the authors provide detailed characterizations on their structures. However, since these compounds are obtained as mixtures, the value of the investigated reaction system on CP preparation is not high. The SCSC transformation is also inadequately studied. In my opinion, this manuscript needs a major revision before it can be published.

Specific comments:

1.       Only one long paragraph is given in the Introduction section, which would inevitably impair the readability of the manuscript. The Introduction section should be divided into several paragraphs.

2.       MOF is the brief of metal-organic framework, instead of “molecular organic framework”.

3.       The maximum yields of the separated compounds should be supplemented to the revised manuscript.

4.       More data about the repeatability, kinetics and temperature-dependence of the interconversion between CP3 and CP4 are necessary to show their potential in humidity sensing. Additionally, its performance should be compared with the presently used humidity sensors.

Author Response

We are re-submitting to you the point-by-point response to the reviewers of the Manuscript ID: polymers-2395563 with title: " Structural and Theoretical Study of Copper(II)-5- Fluoro Uracil Acetate Coordination Compounds: Single-crystal to single-crystal transformation as possible humidity sensor”. Authors: Verónica Garcí1, Andrea García-Hernán,, Fernando Aguilar-Galindo, Josefina Perles and Pilar Amo-Ochoa

This re-submission included the manuscript, supplementary material, and corrected CIFs and checkcifs, as well as responses to the editor and reviewers .

We hope you find our revised version suitable for publication.

Yours sincerely,

Pilar Amo-Ochoa

Point-by-point response to the reviewers

Reviewer 2

In this manuscript, the authors isolate eight compounds from a hydrothermal synthesis system and characterize their structures. This manuscript is generally systematic. Four of the isolated compounds are new, and the authors provide detailed characterizations on their structures. However, since these compounds are obtained as mixtures, the value of the investigated reaction system on CP preparation is not high. The SCSC transformation is also inadequately studied. In my opinion, this manuscript needs a major revision before it can be published.

Specific comments:

Q1. Only one long paragraph is given in the Introduction section, which would inevitably impair the readability of the manuscript. The Introduction section should be divided into several paragraphs.

A1. Following reviewer 2's suggestion, we have proceeded to divide the introduction into several paragraphs.

Q2. MOF is the brief of metal-organic framework, instead of “molecular organic framework”.

A2. Following reviewer 2's suggestion, we have proceeded to correct the specified typo.

Q3. The maximum yields of the separated compounds should be supplemented to the revised manuscript.

A3. The obtained yields of the new compounds 1, CP2, CP3 and CP4 are indicated in the synthesis section of the main text.

Q4. More data about the repeatability, kinetics and temperature-dependence of the interconversion between CP3 and CP4 are necessary to show their potential in humidity sensing. Additionally, its performance should be compared with the presently used humidity sensors.

A4. Following the reviewer's suggestion, we have added in the main text more details about the study of the transformation from CP3 to CP4. We have indicated the number of times that the reversibility study has been carried out, as well as the rate of interconversion. We must indicate that the experiment has been carried out as a proof of concept. Indeed, we have not optimized the synthesis to obtain CP3 with higher yields. By this reason it is not possible to do a more detailed and comparative study with other sensors to see its possible real applicability since it is not the main objective of the manuscript.

Reviewer 3 Report

This manuscript by Verónica García-Vegas, Andrea García-Hernán, Fernando Aguilar-Galindo, Josefina Perles and Pilar Amo-Ochoa reports the synthesis, structural and theoretical study of Copper(II)-5- Fluoro Uracil Acetate Coordination Compounds.

In general, the manuscript is well written and structured, the experimental design appears to be very relevant, and the methodology is appropriate. Furthermore, from the crystallographic point of view, the manuscript is very complete. The article's weakest component is the thorough characterization, as well as the demonstration of the computational calculations' outcomes.

There is no doubt that the work is clearly and well written, and that the information offered to the reader is accurate. I can see from the bibliography that the authors are experts in this topic. However, I would advise the authors to avoid excessive self-citations.

I cannot suggest this work for acceptance unless such following points are improved and clarified:

1. The spectroscopic discussion of the CPs and CC produced was insufficient in terms of characterization. The authors only list the wavelengths detected in the experimental part. FTIR study may help to complete the characterization by describing the spectroscopic differences.

2. For elemental analysis only found % are presented, not the theoretical %. Authors must complete these data.

3. The authors can conduct additional analyses to complete the characterization, which will considerably improve the piece. Among these studies are certain fundamental ones, such as elemental analysis and Powder-RX, which are in the SI file but aren't discussed in the text. I suggest that the authors finish this investigation by quantifying the phases mixture using powder-XRD.

4. The authors place too much emphasis on theoretical calculations, which are even mentioned in the title, but they do not provide data tables, graphs, or comparisons between different basis functions used in molecular orbital calculations that can be used to perform the calculations. From this perspective, the article should be rewritten.

5. The bibliography employed is not updated, only 1 article in 47 is from 2023, and a did from 2020-22 period.

6. Despite outlining or summarizing the research covered in the study, the conclusions do not adequately justify the study's uniqueness. The authors could highlight how they improved on the present state of the art in this section. What is the most significant takeaway from their research?

Minor points:

1. Latin words should usually be printed in italics.

2. Some % signs are far away from the number, but they should be right next to it.

Author Response

Point-by-point response to the reviewers

Reviewer 3

This manuscript by Verónica García-Vegas, Andrea García-Hernán, Fernando Aguilar-Galindo, Josefina Perles and Pilar Amo-Ochoa reports the synthesis, structural and theoretical study of Copper(II)-5- Fluoro Uracil Acetate Coordination Compounds.

In general, the manuscript is well written and structured, the experimental design appears to be very relevant, and the methodology is appropriate. Furthermore, from the crystallographic point of view, the manuscript is very complete. The article's weakest component is the thorough characterization, as well as the demonstration of the computational calculations' outcomes.

There is no doubt that the work is clearly and well written, and that the information offered to the reader is accurate. I can see from the bibliography that the authors are experts in this topic. However, I would advise the authors to avoid excessive self-citations.

I cannot suggest this work for acceptance unless such following points are improved and clarified:

Q1. The spectroscopic discussion of the CPs and CC produced was insufficient in terms of characterization. The authors only list the wavelengths detected in the experimental part. FTIR study may help to complete the characterization by describing the spectroscopic differences.

A1. Following reviewer 3's suggestion, we have proceeded to broaden the discussion of CPs and CCs in terms of characterization by expanding the study of characterization by IR.

Q2. For elemental analysis only found % are presented, not the theoretical %. Authors must complete these data.

A2. Following reviewer 3's suggestion, we have reviewed the elemental analyzes of CP2, CP3 and CP4 and we have verified that they are correct. The calculated elemental analysis is indicated along with the found elemental analysis in parentheses.

Q3. The authors can conduct additional analyses to complete the characterization, which will considerably improve the piece. Among these studies are certain fundamental ones, such as elemental analysis and Powder-RX, which are in the SI file but aren't discussed in the text. I suggest that the authors finish this investigation by quantifying the phases mixture using powder-XRD.

A3. Following reviewer 3's suggestion, we have expanded in the manuscript the discussion of X-ray powder diffractograms and infrared of the obtained  compounds.

Q4. The authors place too much emphasis on theoretical calculations, which are even mentioned in the title, but they do not provide data tables, graphs, or comparisons between different basis functions used in molecular orbital calculations that can be used to perform the calculations. From this perspective, the article should be rewritten.

A4. We thank the referee for his/her comments. Following his/her suggestions, we have included the relative energies of CP3 and CP4 with the two possible approaches for considering the water molecules (isolated or in condensed phase) in a new table (Table 3). Additionally, the theoretical optimized structures are also included in the SI (Figure S15).

Regarding the comparison between basis functions used in molecular orbital calculations, we would like to point out that our calculations are not standard gas-phase molecular calculations, but periodic plane wave-based calculations. 

To ensure accuracy, we used a cut-off energy of 450 eV, exceeding the recommended minimum value of 400 eV for the most electronegative atom (fluorine). We have verified the convergence of our results for this specific energy cut-off. We have clarified the use of plane wave as basis in the computational details section in order to avoid any possible misunderstanding.

Q5. The bibliography employed is not updated, only 1 article in 47 is from 2023, and a did from 2020-22 period.

A5. Following reviewer 3's suggestion, we have revised the bibliography and have introduced more current references in the main text.

A6. Despite outlining or summarizing the research covered in the study, the conclusions do not adequately justify the study's uniqueness. The authors could highlight how they improved on the present state of the art in this section. What is the most significant takeaway from their research?

A6. Following reviewer 3's suggestion, we have revised the conclusions highlighting the greatest significance of this research.

Minor points:

Q7. Latin words should usually be printed in italics.

Q8. Some % signs are far away from the number, but they should be right next to it.

A7 and A8. Following the reviewer's comments, we have proceeded to correct the typographical errors indicated in the current manuscript.

Reviewer 4 Report

The scientific content of this article is referring to the single-crystal to single-crystal transformation in Copper(II)-5- fluoro uracil acetate coordination compounds with potential application as humidity sensors. This study is well-presented and with important potential technological interest. Some minor points that should be examined/ reconsidered by the authors in order to present more appropriately their results.

(a) 2. Materials and Methods: There is an inconsistency on the way mL is written. Keep the same way to all text.

(b) ATR/FTIR analysis: The vibrational characterization of the compounds is an important part and should be discussed in more detail. In particular the FTIR spectra they are not clearly presented and it is not easy to compare them. Revise these Figs. In general, there are a lot of Figs that are included in the Supplementary and they are not discussed in the main ms.  

(c) PXRD.: The authors have compared the experimental and the theoretical XRD of the complexes. This is important for the discussion part. However, the authors are not analyzing these measurements in the main ms.

Author Response

Point-by-point response to the reviewers

Reviewer 4

The scientific content of this article is referring to the single-crystal to single-crystal transformation in Copper(II)-5- fluoro uracil acetate coordination compounds with potential application as humidity sensors. This study is well-presented and with important potential technological interest. Some minor points that should be examined/ reconsidered by the authors in order to present more appropriately their results.

Q1. Materials and Methods: There is an inconsistency on the way mL is written. Keep the same way to all text.

A1. Following reviewer 4's suggestion, we have corrected the inconsistencies related to the expression mL

Q2. ATR/FTIR analysis: The vibrational characterization of the compounds is an important part and should be discussed in more detail. In particular the FTIR spectra they are not clearly presented and it is not easy to compare them. Revise these Figs. In general, there are a lot of Figs that are included in the Supplementary and they are not discussed in the main ms. 

A2. Following the reviewer's suggestion, we have proceeded to include a further discussion of the results obtained by FTIR in the main text. In addition, we have modified the FTIR spectra in the supplementary material by adding the most relevant peaks and comparing the spectra between the different coordination polymers and the starting ligands.

Q3. PXRD.: The authors have compared the experimental and the theoretical XRD of the complexes. This is important for the discussion part. However, the authors are not analyzing these measurements in the main ms.

A3. Following the reviewer's suggestion, we have proceeded to include in the main text a further discussion of the results obtained by powder X-ray diffraction.

Round 2

Reviewer 2 Report

All the concerns have been properly addressed, and this manuscript could be published now. 

Author Response

We are grateful for reviewer 2's comments "ll the concerns have been properly addressed, and this manuscript could be published now". From what we understand that there is no need to make any more changes in the manuscript